# Intermittent Fasting Reduces Neuroinflammation and Cognitive Impairment in High-Fat Diet-Fed Mice by Downregulating Lipocalin-2 and Galectin-3

**DOI:** 10.3390/nu16010159

**Published:** 2024-01-03

**Authors:** Jaewoong Lee, Hyeong Seok An, Hyun Joo Shin, Hye Min Jang, Chae Oh Im, Yeonjun Jeong, Kibaek Eum, Sejeong Yoon, So Jeong Lee, Eun Ae Jeong, Kyung Eun Kim, Gu Seob Roh

**Affiliations:** 1Department of Anatomy and Convergence Medical Science, College of Medicine, Institute of Medical Science, Gyeongsang National University, Jinju 52727, Republic of Korea; woongs1111@gmail.com (J.L.); gudtjr5287@hanmail.net (H.S.A.); k4900@hanmail.net (H.J.S.); gpals759@naver.com (H.M.J.); thwjd5411@naver.com (S.J.L.); jeasky44@naver.com (E.A.J.); kke-jws@hanmail.net (K.E.K.); 2Department of Medicine, College of Medicine, Gyeongsang National University, Jinju 52727, Republic of Korea; atom9998@naver.com (C.O.I.); yak0718@naver.com (Y.J.); eumkb@naver.com (K.E.); cleannclear2769@naver.com (S.Y.)

**Keywords:** intermittent fasting, high-fat diet, lipocalin-2, galectin-3, cognitive impairment

## Abstract

Intermittent fasting (IF), an alternating pattern of dietary restriction, reduces obesity-induced insulin resistance and inflammation. However, the crosstalk between adipose tissue and the hippocampus in diabetic encephalopathy is not fully understood. Here, we investigated the protective effects of IF against neuroinflammation and cognitive impairment in high-fat diet(HFD)-fed mice. Histological analysis revealed that IF reduced crown-like structures and adipocyte apoptosis in the adipose tissue of HFD mice. In addition to circulating lipocalin-2 (LCN2) and galectin-3 (GAL3) levels, IF reduced HFD-induced increases in LCN2- and GAL3-positive macrophages in adipose tissue. IF also improved HFD-induced memory deficits by inhibiting blood–brain barrier breakdown and neuroinflammation. Furthermore, immunofluorescence showed that IF reduced HFD-induced astrocytic LCN2 and microglial GAL3 protein expression in the hippocampus of HFD mice. These findings indicate that HFD-induced adipocyte apoptosis and macrophage infiltration may play a critical role in glial activation and that IF reduces neuroinflammation and cognitive impairment by protecting against blood–brain barrier leakage.

## 1. Introduction

Obesity can lead to non-alcoholic fatty liver disease, type 2 diabetes (T2D), and cognitive impairment [1]. Previous studies indicate that a high-fat diet (HFD) stimulates lipid accumulation in adipose tissue and exacerbates insulin resistance, leading to neuroinflammation [2,3]. Furthermore, increased permeability of the blood–brain barrier (BBB) in obesity causes neuroinflammation and memory deficits [4,5]. However, specific mechanisms that could be exploited for the prevention and treatment of obesity-associated memory deficits are not yet defined.

Lipocalin-2 (LCN2), a neutrophil gelatinase-associated lipocalin, is a secreted protein produced by adipocytes [6]. In obesity, LCN2 promotes the development of insulin resistance and T2D [7,8]. LCN2 is known to increase BBB permeability, allowing harmful substances to enter the brain [9]. However, further research is needed to better understand the role of LCN2 in neuroinflammation and BBB maintenance in obesity.

Galectin-3 (GAL3), a β-galactoside-binding lectin, is expressed by various cell types and regulates innate immune responses [10]. GAL3 promotes inflammation, apoptosis, and oxidative stress in diabetic cardiomyopathy [11]. GAL3 also impairs learning and memory in diabetic patients, leading to mild cognitive impairment [12]. However, the role of GAL3 in HFD-induced inflammation and memory deficits is not fully understood.

Intermittent fasting (IF) is a dietary modification that has recently received attention for its potential to prevent obesity and T2D [13]. There is growing evidence that chronic IF benefits cognitive function in HFD-fed mice [14]. Interestingly, IF for 20 weeks increases hippocampal neuron tolerance to excitotoxic stress in mice, suggesting a neuroprotective effect [15]. However, the beneficial effects of long-term IF on HFD-induced neuroinflammation and memory deficits have yet to be fully examined.

Given the common roles of LCN2 and GAL3 in the chronic inflammation associated with obesity and T2D, the functional relationships between adipose tissue macrophage-derived LCN2 and GAL3 and neuroinflammation would be expected in the diabetic brain with memory deficits. We previously showed that LCN2 has an inflammatory role in the diabetic brain with BBB leakage [4]. However, the exact mechanisms of IF on LCN2 and GAL3-mediated adipose tissue macrophage infiltration and neuroinflammation have not been fully studied. The present study aimed to investigate the protective role of chronic IF against HFD-induced inflammation and associated diabetic encephalopathy. In addition, we examined the effects of IF on the crosstalk between adipocyte death-related macrophage accumulation and hippocampal inflammation in HFD-fed mice with BBB leakage. We report that IF attenuated HFD-induced inflammation and memory deficits by downregulating LCN2 and GAL3 proteins.

## 2. Materials and Methods

### 2.1. Animals and IF Mouse Model

Three-week-old male C57BL/6 mice were purchased from KOATECH (Pyeongtaek, Republic of Korea). The mice were divided into normal diet (ND; *n* = 10), HFD (*n* = 10, 60% kcal from fat, Research Diets Inc., New Brunswick, NJ, USA), and HFD + IF (HIF; *n* = 12) groups. Mice in the ND and HFD groups were fed a ND or HFD for 30 weeks, whereas mice in the HIF group were fed an HFD for 8 weeks and then switched to an IF protocol consisting of alternating 24 h periods of fasting and feeding for 22 weeks (Appendix A). We measured food intake and energy intake every other day for 16 weeks at 12 weeks after IF protocol (Appendix A). All mice fasted overnight before sacrifice at 34 weeks of age. Mice were individually housed under an alternating 12 h light/dark cycle.

### 2.2. Echo MRI

EchoMRI (Whole Body Composition Analyzer, Houston, TX, USA) was performed on mice to quantify body fat mass.

### 2.3. Glucose Tolerance Test (GTT) and Insulin Tolerance Test (ITT)

GTT and ITT were performed as previously described [16] using D-glucose (2 g/kg, Sigma-Aldrich, St. Louis, MO, USA) or insulin (0.75 U/kg, Humulin-R, Eli Lilly, Indianapolis, IN, USA). After intraperitoneal injection of D-glucose or insulin, blood samples were obtained from tail vein. The glucose levels from GTT and ITT were determined using an Accu-Chek glucometer (Roche Diagnostics GmbH, Mannheim, Germany).

### 2.4. Enzyme-Linked Immunosorbent Assay (ELISA)

Serum protein levels were measured using mouse LCN2 (R&D Systems, Minneapolis, MN, USA), GAL3 (Abcam, Cambridge, UK), and matrix metalloproteinase 9 (MMP9; R&D Systems) enzyme-linked immunosorbent assay (ELISA) kits according to the manufacturers’ protocols.

### 2.5. Hematoxylin and Eosin (H&E) Staining

WATs (*n* = 3–4 mice per group) were fixed in 4% paraformaldehyde for 12 h at 4 °C. Samples were embedded in paraffin, cut into 5-μm sections, stained with hematoxylin and eosin (H&E; Abcam, Cambridge, MA, USA), and visualized under BX53 light microscopy (Olympus, Tokyo, Japan). The number of crown-like structures (CLSs) were counted in three randomly selected fields.

### 2.6. Terminal Deoxynucleotidyl Transferase Dutp Nick end Labeling (TUNEL) Assay

TUNEL assay was used to measure the degree of apoptosis in WATs using an in situ cell death detection kit (Roche Molecular Biochemicals, Mannheim, Germany) according to the manufacturer’s protocol. For the counting of the number of TUNEL-positive cells (*n* = 3–4 mice per group), three fields (400 μm × 400 μm) were randomly selected from each section using ImageJ software (Version 1.52a, NIH, Bethesda, MD, USA).

### 2.7. Western Blot Analysis

Frozen WATs and hippocampi (*n* = 3–4 mice per group) were homogenized in T-PER lysis buffer (Thermo Fisher Scientific, Carlsbad, CA, USA) with a protease and phosphatase inhibitor cocktail (Thermo Fisher Scientific). After bicinchoninic acid assay (Thermo Fisher Scientific) for protein concentration, proteins were loaded and electroblotted. Blots were probed with primary antibodies (Appendix A). α-tubulin and β-actin were used as internal controls to normalize protein content in tissue samples. Protein bands were detected using enhanced chemiluminescence substrates (Pierce, Rockford, IL, USA), and chemiluminescence was analyzed using an LAS-4000 instrument (Fujifilm, Tokyo, Japan). The Multi-Gauge V 3.0 image analysis program (Fujifilm, Tokyo, Japan) was used for densitometry analysis.

### 2.8. Double or Triple Immunofluorescences

Sections of deparaffinized WATs and frozen brains (*n* = 3–4 mice per group) were incubated with 5% serum for 1 h at room temperature followed by incubation with primary antibodies (Appendix A). After washing three times, sections were incubated with corresponding Alexa Fluor 488-, 594-, or 680-conjugated secondary antibody (Invitrogen, Carlsbad, CA, USA). Nuclei were counterstained with 4′,6-diamidino-2-phenylindole (DAPI; Invitrogen). Slides were mounted with VectaMount (Vector Laboratories, Burlingame, CA, USA), and representative images were taken using an FV3000 microscope (Olympus, Tokyo, Japan). For the counting of the number of extravascular albumin from hippocampal section, three fields (200 μm × 200 μm) were randomly selected from each section using ImageJ software (Version 1.52a, NIH).

### 2.9. Morris Water Maze (MWM)

For 5 days, the MWM test was conducted as previously described [16]. Mice (*n* = 7 mice per group) received four daily trials for four consecutive days. A video-tracking program (Noldus EthoVision XT7, Noldus Information Technology, Wageningen, The Netherlands) recorded latency to find the platform. On the last day, the platform was removed, and the numbers of crossings in the target quadrant (i.e., where the platform had been located) and the platform area were analyzed.

### 2.10. Statistical Analysis

Statistical analyses were performed using PRISM 7.0 (GraphPad Software Inc., San Diego, CA, USA). Group differences were determined by one-way analysis of variance (ANOVA) followed by post hoc analysis with Tukey’s tests. All values are expressed as mean ± standard error of the mean (SEM). A *p*-value < 0.05 was considered statistically significant.

## 3. Results

### 3.1. IF Attenuates Adipocyte Death and Macrophage Infiltration in the WAT of HFD Mice

To establish an experimental diabetic mouse model, mice were fed an HFD for 30 weeks. To examine the impact of IF on HFD mice, HIF mice were fed an HFD for 8 weeks and then switched to an IF protocol consisting of alternating 24 h periods of fasting and feeding for 22 weeks (Appendix A). Compared with mice fed an ND, HFD mice had heavier body weights and greater fat mass, whereas IF reversed these changes (Body weight; F (2, 29) = 42.67, *p* < 0.0001, Fat mass; F (2, 29) = 54.51, *p* < 0.0001) (Figure 1A,B). Glucose and insulin tolerance tests showed that HFD mice had impaired glucose tolerance compared with ND mice. However, IF significantly attenuated HFD-induced insulin resistance (GTT 0 min; F (2, 29) = 10.66, *p* = 0.0003, GTT 30 min; F (2, 29) = 1.818, *p* = 0.1804, GTT 60 min; F (2, 29) = 1.357, *p* = 0.2732, GTT 90 min; F (2, 29) = 2.748, *p* = 0.0807, GTT 120 min; F (2, 29) = 4.471, *p* = 0.0203) (ITT 0 min; F (2, 29) = 6.894, *p* = 0.0036, ITT 15 min; F (2, 29) = 10.46, *p* = 0.0004, ITT 30 min; F (2, 29) = 19.52, *p* < 0.0001, ITT 45 min; F (2, 29) = 17.52, *p* < 0.0001, ITT 60 min; F (2, 29) = 5.307, *p* = 0.0109) (Figure 1C,D). To investigate the effects of IF on HFD-induced WAT inflammation, we first examined adipocyte death and macrophage infiltration. Histological analysis revealed that HFD mice had many CLSs and TUNEL-positive cells in the WAT, whereas IF significantly reversed these changes (CLSs; F (2, 26) = 15.70, *p* < 0.0001, TUNEL; F (2, 17) = 16.94, *p* < 0.0001) (Figure 1E–G). Additionally, the HFD-induced increase in the ratio of Bax-to-Bcl-2 protein expression was reduced by IF (F (2, 8) = 6.398, *p* = 0.0219) (Figure 1H). We next assessed macrophage infiltration in HFD-induced WAT apoptosis. Double immunofluorescence analysis revealed that IF attenuated the infiltration of F4/80-positive macrophages into perilipin-1-free adipocytes in HFD mice (Figure 1I). Together, these findings indicate that IF may improve insulin resistance in HFD mice by reducing adipocyte death and macrophage infiltration.

### 3.2. IF Reduces Circulating and WAT LCN2 Protein Levels in HFD Mice

Because LCN2, an adipocytokine, is closely related to adipocyte death and inflammation in obesity [17,18], we evaluated the effects of IF on LCN2 protein levels in HFD mice. IF attenuated the increased serum LCN2 level in HFD mice (F (2, 15) = 21.39, *p* < 0.0001) (Figure 2A). In WAT, the HFD-induced increase in LCN2 protein level was significantly reduced by IF (F (2, 8) = 7.867, *p* = 0.0129) (Figure 2B). Triple immunofluorescence showed the presence of LCN2-positive cells in myeloperoxidase-positive neutrophils and F4/80-positive macrophages in the WAT of HFD mice. However, these LCN2-positive neutrophils and macrophages were not observed in HIF mice (Figure 2C). These results suggest that IF reduces LCN2-mediated inflammation in HFD mice.

### 3.3. IF Reduces Circulating and WAT GAL3 Protein Levels in HFD Mice

Because GAL3 is related to adipocyte inflammation in HFD-induced obesity [19], we evaluated the effects of IF on GAL3 protein levels in HFD mice. Consistent with serum LCN2 levels, IF inhibited the increase in GAL3 levels in HFD mice (F (2, 18) = 34.11, *p* < 0.0001) (Figure 3A). GAL3 protein expression was elevated in the WAT of HFD mice compared with ND mice, whereas IF significantly reversed this increase (F (2, 8) = 20.99, *p* = 0.0007) (Figure 3B). Double immunofluorescence showed the presence of many GAL3-positive cells in LCN2-positive cells in HFD mice (Figure 3C). However, these colocalized cells were reduced by IF. Thus, these findings indicate that IF attenuates circulating and macrophage-derived GAL3 protein in HFD mice.

### 3.4. IF Improves Memory Deficits in HFD Mice

To assess whether IF improves memory deficits in HFD mice, we performed the MWM test. ND and HIF mice tended to show shorter escape latencies across training for 4 consecutive days. However, on the last day, there was significant decrease in escape latency in HIF mice compared with HFD mice (Training on the first day; F (2, 81) = 2.106, *p* = 0.1283, Training on the second day; F (2, 81) = 4.126, *p* = 0.0197, Training on the third day; F (2, 81) = 0.3548, *p* = 0.7024, Training on the last day; F (2, 18) = 6.176, *p* = 0.0091) (Figure 4A). In addition, we found that there were no significant differences in swimming distance or speed among groups on the last day (swimming distance; F (2, 18) = 2.410, *p* = 0.1182, swimming speed; F (2, 18) = 2.231, *p* = 0.1363) (Figure 4B,C). Observation of swimming routes on the test day showed that HIF mice exhibited more of a spatial bias toward the former platform location compared with HFD mice (Figure 4D). Moreover, IF increased the numbers of crossings in the target quadrant and target zone compared with HFD mice (Numbers of crossings in the target quadrant; F (2, 18) = 6.176, *p* = 0.0091, Numbers of crossings in the target zone; F (2, 18) = 8.222, *p* = 0.0029) (Figure 4E,F), indicating that IF improves HFD-induced memory deficits.

### 3.5. IF Inhibits BBB Leakage in the Hippocampus of HFD Mice

Given the important role of the hippocampus in memory function, vascular abnormalities in the hippocampus are likely to be causally related to cognitive impairment in diabetic patients [5]. To investigate whether the improvement in memory deficits by IF was due to protection against BBB leakage, we measured BBB-related protein levels (Figure 5A). Hippocampal levels of claudin-5 were markedly lower in HIF mice compared with HFD mice. By contrast, zonula occludens-1 (ZO-1) expression was lower in HFD mice, whereas IF restored its expression (claudin-5; F (2, 8) = 21.36, *p* = 0.0006, ZO-1; F (2, 8) = 9.555, *p* = 0.0076) (Figure 5A). Furthermore, HFD-induced intercellular adhesion molecule-1 (ICAM-1), a leukocyte adhesion receptor on endothelial cell membranes, and MMP9 were significantly reduced in the hippocampus of HIF mice (ICAM-1; F (2, 8) = 18.6, *p* = 0.0010, MMP9; F (2, 8) = 43.23, *p* < 0.0001) (Figure 5A). We also confirmed that an increase in circulating MMP9 level in HFD mice was decreased by IF (F (2, 15) = 10.75, *p* = 0.0013) (Figure 5B). Next, we performed immunofluorescence staining with aquaporin-4 (AQP4) and albumin to detect albumin in extravascular regions. AQP4 is a specific component of astroglial endfeet that comprises the BBB in close contact with endothelial cells [20,21]. We observed many albumins near AQP4-positive astrocytes in the hippocampus of HFD mice compared with ND mice (Figure 5C,D). However, IF significantly alleviated extravascular albumins in the hippocampus of HFD mice (F (2, 33) = 37.20, *p* < 0.0001). Thus, these results suggest that IF protects against BBB leakage and inhibits hippocampal inflammation in HFD mice.

### 3.6. IF Reduces Microglial GAL3 and Astrocytic LCN2 in the Hippocampus of HFD Mice

Given that levels of GAL3 and LCN2 proteins were increased in the serum and WAT of HFD mice, we investigated the effect of IF on microglial GAL3 and astrocytic LCN2 expression in HFD mice. Western blot analysis revealed that IF significantly reduced HFD-induced increases in hippocampal GAL3 and LCN2 protein levels (GAL3; F (2, 8) = 35.53, *p* = 0.0001, LCN2; F (2, 8) = 18.67, *p* = 0.0010) (Figure 6A). Double immunofluorescence revealed that GAL3-positive Iba-1 microglia mainly localized around vascular regions in the hippocampus of HFD mice, but GAL3-positive microglia were not observed in IF mice (Figure 6B). In addition, LCN2 colocalized with GFAP-stained vessels in HFD mice, but these colocalized cells were not observed in HIF mice (Figure 6B). These findings suggest that IF mitigates BBB permeability-induced glial activation in HFD mice by downregulating microglial GAL3 and astrocytic LCN2 proteins.

### 3.7. IF Reduces Hippocampal Inflammation in HFD Mice

To verify whether glial activation is linked to neuroinflammation in HFD mice, we determined the effect of IF on inflammation-related proteins in the hippocampus of HFD mice. Western blot analysis showed that tumor necrosis factor-α (TNF-α) and TNF receptor-1 (TNFR1) levels in the hippocampus of HFD mice were reduced by IF (TNF-α; F (2, 8) = 8.103, *p* = 0.0119, TNFR1; F (2, 8) = 16.06, *p* = 0.0016) (Figure 7A,B). Consistently, hippocampal interleukin-6 (IL-6) levels were significantly lower in HIF mice than in HFD mice (F (2, 8) = 6.673, *p* = 0.0197) (Figure 7A,B). Hrigh-mobility group box-1 (HMGB1) is endocytosed through the receptor for advanced glycation end products (RAGE), which is associated with inflammatory diseases [22]. Western blot analysis showed that IF markedly reduced hippocampal HMGB1 and RAGE expression in HFD mice (HMGB1; F (2, 8) = 11.67, *p* = 0.0042, RAGE; F (2, 8) = 22.66, *p* = 0.0005) (Figure 7A,B). These results indicate that IF attenuated HFD-induced hippocampal inflammation.

## 4. Discussion

We investigated the effects of IF on the crosstalk between adipocyte death-related macrophage accumulation and hippocampal inflammation in HFD mice. First, we found that IF attenuated HFD-induced adipocyte death and macrophage infiltration in the WAT and improved insulin resistance. Second, IF reduced LCN2- and GAL3-positive macrophages in the WAT of HFD mice as well as serum LCN2 and GAL3 levels. Third, IF attenuated HFD-induced BBB leakage, neuroinflammation, and memory deficits. Finally, IF reduced astrocytic LCN2 and microglial GAL3 expression in the hippocampus of HFD mice. Thus, these findings suggest that IF could be used as drug replacement therapy to improve cognitive impairment resulting from HFD-induced neuroinflammation and BBB breakdown.

A critical finding of this study is that interrupting the HFD regimen with IF caused dramatic weight loss and attenuated insulin resistance and adipocyte death. Moreover, the decrease in CLSs, which are characteristic of WAT apoptosis and macrophage infiltration [23,24], suggests that metabolic dysfunction or insulin resistance can be corrected by caloric reduction. These results are consistent with our previous study, which showed that caloric restriction (2 g/day) reduced body weight and adipocyte macrophage infiltration in HFD mice [5]. Increased adipocyte death increases the activation of proinflammatory adipocyte resident macrophages or other immune cells, thereby aggravating WAT inflammation [25,26]. Thus, mitigating WAT inflammation is crucial for protecting against obesity-associated metabolic disorders. In the present study, IF reduced histological CLS and macrophage infiltration in the WAT of HFD mice. These results indicate that IF, an alternative to continuous caloric restriction, may attenuate WAT inflammation by inhibiting adipocyte death and macrophage infiltration and improving insulin resistance in HFD mice.

LCN2 is an adipocytokine that is abundantly secreted from adipose tissue, including neutrophils and macrophages [6,27]. LCN2 is a major contributor to inflammation in adipose and other tissue in HFD-induced obesity [27,28]. Our previous studies demonstrate that LCN2 plays an inflammatory role in the WAT, liver, and hippocampus in animal models of metabolic disorders, including obesity, non-alcoholic fatty liver disease, and T2D [4,8,18]. As expected, in the present study, we found that in addition to reducing circulating LCN2 levels, IF reduced HFD-induced LCN2 expression and LCN2-positive neutrophils and macrophages in the WAT. These results indicate that IF may alleviate HFD-induced WAT inflammation by suppressing LCN2 expression.

GAL3, a lectin mainly secreted by macrophages, promotes inflammation and insulin insensitivity in obesity [29]. Consistent with evidence that GAL3 is increased in adipose tissue macrophages in HFD mice [19], we found that IF reduced the HFD increase in serum GAL3 and WAT protein levels and co-localization with LCN2-positive macrophages. As IF may exert an anti-inflammatory effect by triggering the transition of M2 phenotype macrophages in mice [30], these findings suggest that IF reverses HFD-induced macrophage activation in the WAT by downregulating GAL3-mediated macrophage infiltration.

Given the critical role of the hippocampus in learning and memory, vascular abnormalities within the hippocampus may be causally related to cognitive impairment in patients with diabetes or Alzheimer’s disease [5,31]. The hippocampal BBB is more susceptible to disruption than other brain regions, thereby increasing the impact on learning and memory [32]. Thus, BBB breakdown is considered an early biomarker of human cognitive impairment [33]. It is widely reported that elevated levels of inflammatory mediators promote neuroinflammation by triggering detrimental neutrophil/microglia activation in the diabetic brain through BBB leakage [34,35]. We hypothesize that HFD-induced increases in circulating LCN2 and GAL3 invade the leaky BBB and activate glia, resulting in local neuroinflammation. Activated microglia can secrete high levels of TNF-α, which exacerbates neuroinflammation and further potentiates BBB permeability, directly impairing neuronal network dynamics and causing memory deficits [36]. Consistent with previous studies [5,37], we found that IF reversed the HFD-induced increases in proinflammatory cytokines (TNF-α and IL-6), TNFR1, HMGB1, and RAGE in the hippocampus. TNF-α is known to increase BBB leakage via degradation and downregulation of BBB tight junction proteins [37]. In HFD/streptozotocin-treated diabetic mice with memory deficits, we previously demonstrated that ZO-1 protein is reduced in the hippocampus, and many neutrophils are present in the extravascular space of vessels ensheathed by GFAP-expressing astrocytes [4]. In the present study, HFD mice exhibited upregulation of claudin-5, ICAM-1, and MMP9 expression in the hippocampus, whereas levels of hippocampal ZO-1 protein were reduced. However, these changes were reversed by IF. Furthermore, the HFD-induced increase in the amount of extravascular albumin was reversed by IF, suggesting that IF protects against HFD-induced BBB leakage. Previous studies report that increased MMP9 secretion and protein levels in human brain microvascular endothelial cells are attenuated by an MMP inhibitor [38]. In diabetic mice, MMPs promote BBB disruption and neuroinflammation [39]. These findings suggest that the hippocampal BBB plays an important role in diabetes-related memory deficits and that BBB components may represent novel therapeutic targets for mitigating the neuroinflammation and cognitive impairment associated with chronic low-grade inflammatory conditions, including obesity and diabetes.

Our previous studies report that upregulating systemic and hippocampal LCN2 levels causes cognitive deficits in HFD or ob/ob mice [40,41]. Conversely, LCN2 deficiency improves insulin resistance and diabetic encephalopathy [7,42]. In accordance with evidence that hippocampal LCN2 levels are increased in diabetic encephalopathy [4,7], we found that serum and hippocampal LCN2 levels were elevated in HFD-fed mice and normalized by IF. A previous study shows that LCN2 released from astrocytes activates microglia, further aggravating diabetes-induced neuroinflammation [43]. This supports the notion that LCN2 deficiency reduces levels of reactive gliosis and inflammatory cytokines in mice with diabetic encephalopathy. Based on the present findings, we suggest that LCN2 crosses the BBB as an inflammatory mediator and activates LCN2-positive astrocytes and GAL3-positive microglia in the diabetic brain. The increased immune response of glial cells may be caused by direct neuronal damage of a HFD or secreted LCN2 and GAL3 proteins caused by BBB damage. Consistent with evidence that serum and hippocampus GAL3 levels are associated with neuroinflammation and memory deficits in diabetic rats [12], we also found increased levels of LCN2 and GAL3 in the serum and hippocampus of HFD mice with memory deficits. Therefore, these results strongly suggest that LCN2 and GAL3 are systemic inflammatory mediators that are causative of diabetic neuroinflammation. However, IF was capable of attenuating the accumulation of LCN2- and GAL3-positive cells through BBB leakage in the hippocampus of HFD mice. Although we did not observe microglial expression of LCN2 in HFD mice, HFD mice had more GAL3-positive microglia in the hippocampus than ND mice, and this effect was reversed by IF. There is substantial evidence that GAL3 contributes to inflammation, microglial activation, and neurodegeneration. Recent studies also suggest that GAL3 is a key player in microglia-mediated neuroinflammation [44,45]. Therefore, our results suggest that IF provides neuroprotection against the aggravating neuroinflammatory response in diabetic encephalopathy by downregulating astrocytic LCN2 and microglial GAL3 levels.

## 5. Conclusions

We demonstrated that LCN2 and GAL3, as proinflammatory mediators, exert local effects on systemic and neuroinflammation via BBB leakage under an HFD-induced diabetic condition. Systemic LCN2- and GAL3-mediated WAT macrophages induced hippocampal inflammation via BBB leakage, but these pathological findings were reversed by IF. However, this study also has some limitations. First, it did not provide evidence for how specific immune cells are regulated by GAL3 in diabetic encephalopathy. Second, cellular crosstalk between the WAT and hippocampus must be further investigated. Third, it remains to be determined whether myeloid-specific GAL3 deficiency plays an anti-inflammatory role in the diabetic brain. Thus, further studies will help establish the precise mechanisms by which IF protects against cognitive impairment associated with metabolic disorders such as obesity and T2D.

## Figures and Tables

**Figure 1 nutrients-16-00159-f001:**
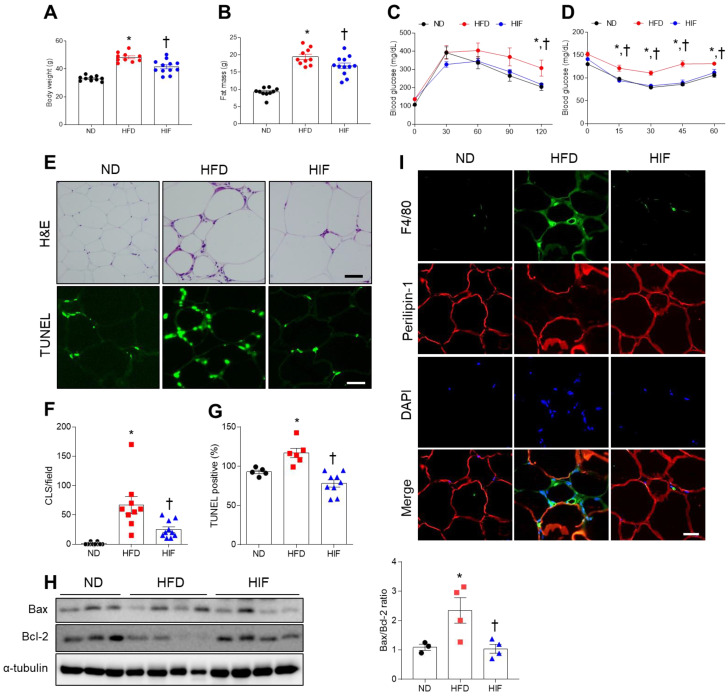
Effects of IF on insulin resistance and adipocyte death in the WAT of HFD mice. (**A**) Body weight (*n* = 10–12). (**B**) Fat mass (*n* = 10–12). (**C**) Glucose tolerance test (*n* = 10–12). (**D**) Insulin tolerance test (*n* = 10–12). (**E**) Representative images of H&E and TUNEL staining in WAT sections (*n* = 3–4). Scale bar, 100 μm. (**F**) Quantification of CLSs in H&E-stained sections. (**G**) Quantification of TUNEL-positive cells in TUNEL-stained sections. (**H**) Western blot analysis of Bcl-2 and Bax proteins in WAT lysates (*n* = 3–4). Quantification of Bax-to-Bcl-2 ratio. (**I**) Representative images of double immunofluorescence staining of F4/80 (green) and perilipin-1 (red) in WAT sections. Nuclei were stained with DAPI. Scale bar, 100 µm. Significance was determined by one-way ANOVA. * *p* < 0.05 versus ND. ^†^
*p* < 0.05 versus HFD.

**Figure 2 nutrients-16-00159-f002:**
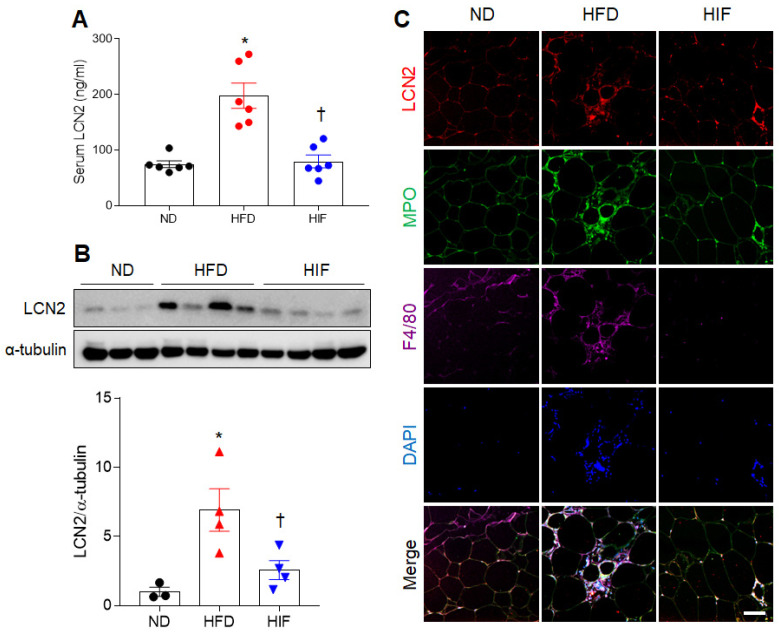
Effects of IF on serum and WAT LCN2 protein levels in HFD mice. (**A**) Serum LCN2 levels (*n* = 6) as assessed using ELISA. (**B**) Western blot and quantitative analysis of LCN2 protein in WAT lysates (*n* = 3–4). Protein levels were normalized to α-tubulin from the same immunoblot. (**C**) Representative images of triple immunofluorescence staining of LCN2 (red), MPO (green), and F4/80 (purple) in WAT sections. Nuclei were stained with DAPI. Scale bar, 50 µm. Significance was determined by one-way ANOVA. * *p* < 0.05 versus ND. ^†^
*p* < 0.05 versus HFD.

**Figure 3 nutrients-16-00159-f003:**
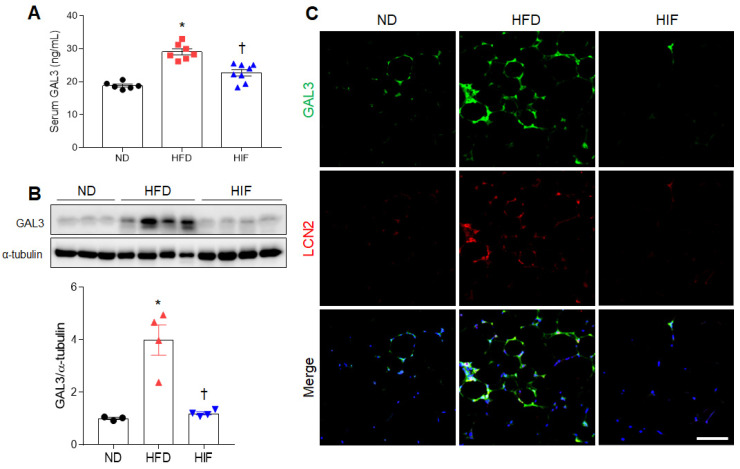
Effects of IF on serum and WAT GAL3 protein levels in HFD mice. (**A**) Serum GAL3 levels (*n* = 6–8) as assessed using ELISA. (**B**) Western blot and quantitative analysis of GAL3 protein in WAT lysates (*n* = 3–4). Protein levels were normalized to α-tubulin from the same immunoblot. (**C**) Representative images of GAL3 (green) and LCN2 (red) double immunofluorescence staining in WAT sections. Nuclei were stained with DAPI. Scale bar, 50 µm. Significance was determined by one-way ANOVA. * *p* < 0.05 versus ND. ^†^
*p* < 0.05 versus HFD.

**Figure 4 nutrients-16-00159-f004:**
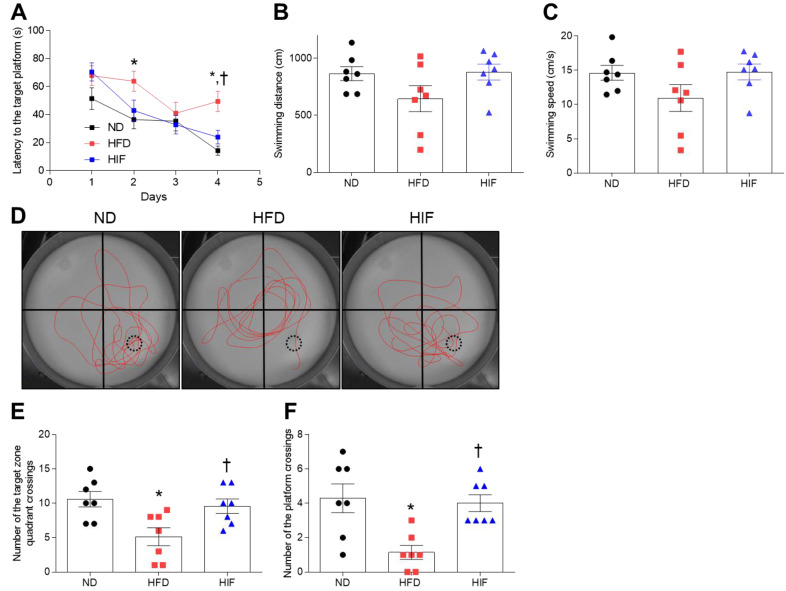
Effect of IF on cognitive impairment in HFD mice. (**A**) Latency to reach the target platform over 4 days of MWM training (*n* = 7). (**B**,**C**) Swimming speed (**B**) and swimming distance (**C**) on the test day (*n* = 7). (**D**) Representative images of swimming paths without the platform during testing. Red lines indicate the swimming path. Numbers of target zone quadrant (**E**) and platform (**F**) crossings on the test day (*n* = 7). Black dashed circles indicate the location of the hidden platform. Significance was determined by one-way ANOVA. * *p* < 0.05 versus ND. ^†^
*p* < 0.05 versus HFD.

**Figure 5 nutrients-16-00159-f005:**
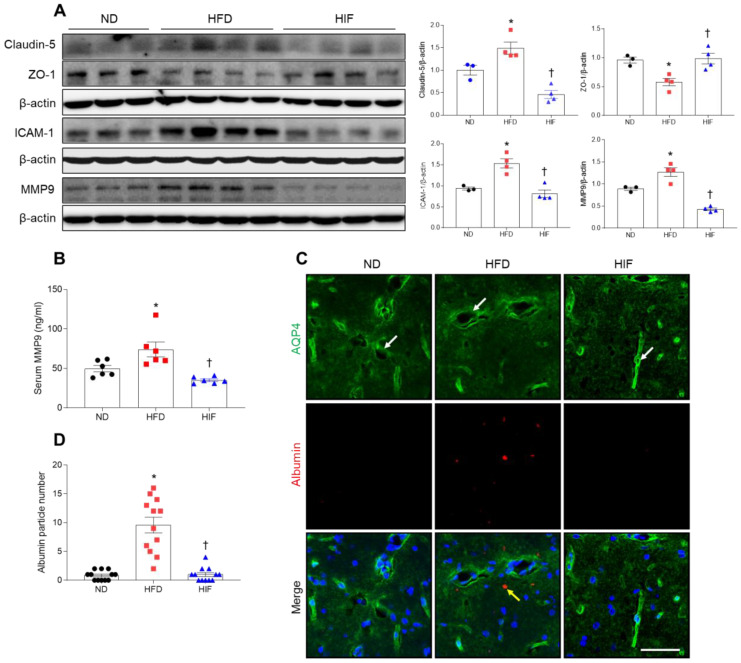
Effect of IF on BBB leakage in the hippocampus of HFD mice. (**A**) Western blot and quantitative analysis of claudin-5, ZO-1, ICAM-1, and MMP9 in hippocampal lysates (*n* = 3–4). Protein levels were normalized to β-actin from the same immunoblot. (**B**) Serum MMP9 level (*n* = 6). (**C**) Representative images of double immunofluorescence staining of AQP4 (green) and albumin (red) in hippocampal sections. White arrows indicate AQP4-positive astroglial endfeet. Yellow arrow indicates extravascular albumin. Nuclei were stained with DAPI. Scale bar, 50 µm. (**D**) Quantitative analysis of albumin from (**C**). Significance was determined by one-way ANOVA. * *p* < 0.05 versus ND. ^†^
*p* < 0.05 versus HFD.

**Figure 6 nutrients-16-00159-f006:**
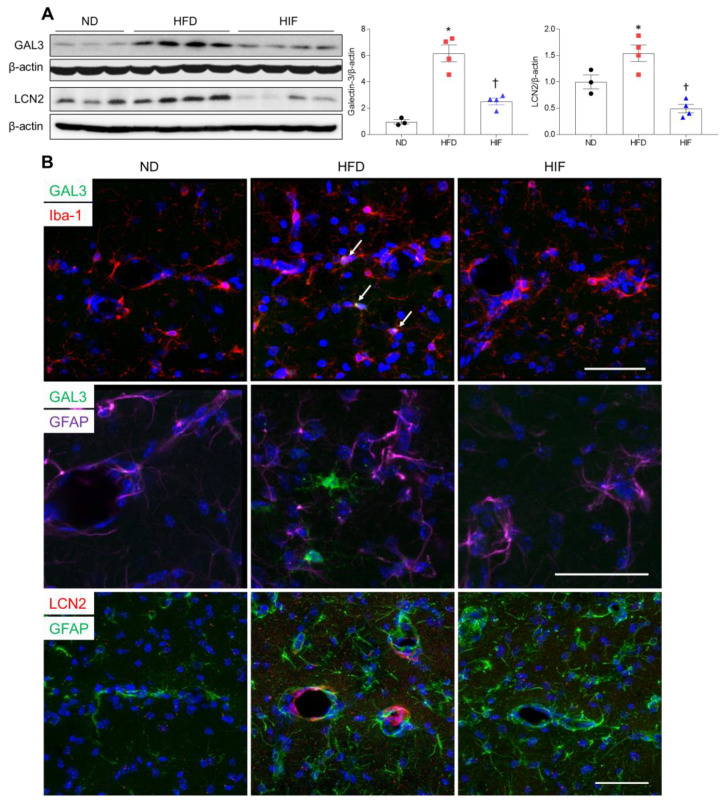
Effects of IF on GAL3 and LCN2 protein in the hippocampus of HFD mice. (**A**) Western blot and quantitative analysis of GAL3 and LCN2 in hippocampal lysates (*n* = 3–4). Protein levels were normalized to β-actin. (**B**) Representative images of double immunofluorescence staining of GAL3 (green) and Iba-1 (red), GAL3 (green) and GFAP (purple), or LCN2 (red) and GFAP (green) in hippocampal sections. Arrows indicate co-localized GAL3 and Iba-1-positive microglia. Nuclei were stained with DAPI. Scale bar, 50 µm. Significance was determined by one-way ANOVA. * *p* < 0.05 versus ND. ^†^
*p* < 0.05 versus HFD.

**Figure 7 nutrients-16-00159-f007:**
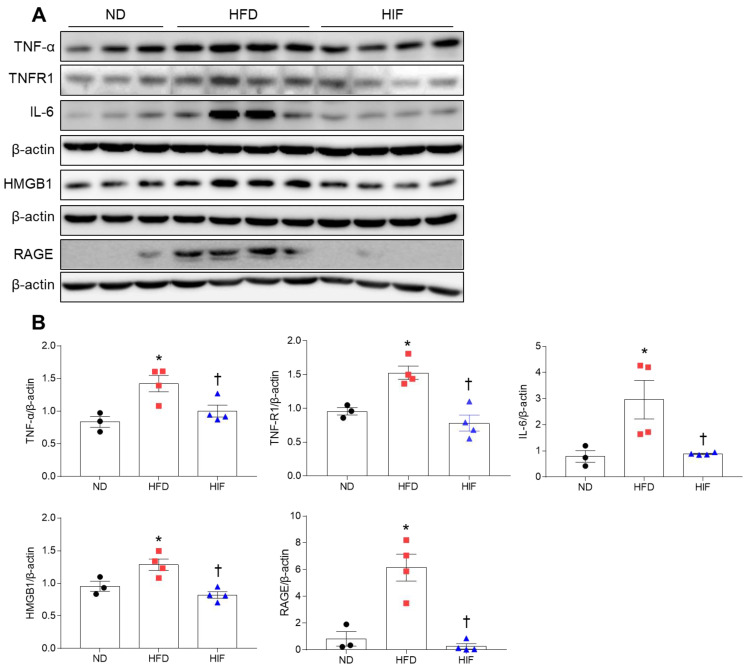
Effects of IF on neuroinflammation in the hippocampus of HFD mice. (**A**,**B**) Western blot (**A**) and quantitative analysis (**B**) of TNF-α, TNFR1, IL-6, HMGB1, and RAGE in hippocampal lysates (*n* = 3–4). Protein levels were normalized to β-actin from the same immunoblot. Significance was determined by one-way ANOVA. * *p* < 0.05 versus ND. ^†^
*p* < 0.05 versus HFD.

## Data Availability

The data presented in this study are available in this manuscript.

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
