# Peer review of "Intermittent Fasting Reduces Neuroinflammation and Cognitive Impairment in High-Fat Diet-Fed Mice by Downregulating Lipocalin-2 and Galectin-3"

_nutrients, 2024, doi:10.3390/nu16010159_

Round 1
Reviewer 1 Report
Comments and Suggestions for Authors
The paper entitled “Intermittent fasting reduces neuroinflammation and cognitive impairment in high-fat diet-fed mice by downregulating lipocalin-2 and galectin-3” found that HFD-induced adipocyte apoptosis and macrophage infiltration may play a critical role in glial activation and that IF reduces neuroinflammation and cognitive impairment by protecting against blood-brain barrier leakage. To my knowledge, there were already many researches confirmed that IF could reduce neuroinflammation and alleviate cognitive impairment in high-fat diet (HFD)-fed mice (Nutrients, 2020,13(1), 10; Nutrients, 2021,13(9), 3166; Medical journal of the Islamic Republic of Iran, 2022,36, 12). However, there are numerous issues needed to be further solved in the study.
1. The authors do not establish a clear and logical relationship between neuroinflammation, cognitive impairment and lipocalin-2 and galectin-3.
2. The Graphical abstract is missing.
3. Reference 12 focuses on the diabetic rats tests given and does not mention diabetic patients. Please check that the references are correct and match up.
4. Detailed replicates about each indicator should be included in figure notes.
5. Important information such as food intake and energy intake is missing.
6. There is no quantitative analysis of the immunofluorescence staining reported in this paper.
7. This part of the subtitle (3.2 and 3.3) cannot summarize the content of the Figure 2 and Figure 3.
8. In Figure 4, especially the poor reproducibility of the ethology is frightening, and there were only 7 isolates tested, not 10-12.
9. In Figure 5A, the quantitative analysis result of Claudin-5/β-actin is wrong. Why the replicates in Group HIF is 3 not 4? In Figure 6A, the quantitative analysis result of LCN2/β-actin is wrong. Why the replicates in Group HIF is 5 not 4? In Figure 7B, the quantitative analysis result of TNF-R1/β-actin is wrong. Why the replicates in Group HIF is 3 not 4? Through what measures did you ensure the precision and authenticity of your data?
10. The results in particular should include more numerical details to describe the results (comparisons, P values, etc.)
Comments on the Quality of English LanguageMinor editing of English language required.
Reviewer 2 Report
Comments and Suggestions for Authors
In humans obesity and type 2 diabetes are associated with cognitive decline, which could be associated with peripheral inflammation affecting the central nervous system. Intermittent fasting has been shown to have beneficial effects on the various pathological effects of obesity in humans and animal models, although the evidence for its impact on cognitive function in humans remains limited but of considerable interest.
The study described by Lee et al. aims to investigate the effects of intermittent fasting (IF) on diet induced memory deficit in male mice. The authors use a conventional approach to induce obesity in mice providing them with either normal diet or high fat diet (HFD) from an age of 3-4 weeks. After 8 weeks of HFD the intermittent fasted group were started on an IF protocol. The experiment was stopped at 34 weeks for all mice. The initial results showed that, as expected, the HFD mice had developed glucose resistance and insulin intolerance, their adipose tissue was infiltrated with pro-inflammatory macrophages. Importantly HFD mice had developed memory deficits as measured using a Morris water maze, which is commonly used for assessing learning and memory in mice, and particularly of hippocampal function. The authors find that IF reverses/prevents the impact of HFD feeding not only on metabolic and inflammatory endpoints, but also improves memory deficit. Using immunofluorescence they show this may be related to improved blood brain barrier (BBB) function in the hippocampus (using aquaporin4 and albumen as a markers). Furthermore, they show that IF reduced astrocytic lipocalin-2 and microglial galectin-3 positive cells in the hippocampus, suggesting a beneficial effect against neuroinflammation in this region of the brain. These data provide some mechanistic insight into how IF may be preventing memory deficit in the HFD fed mice.
The study has been well described and performed, the methods are appropriate and relatively extensive data from a number of different relevant parameters/endpoints are provided. Based on this I feel the conclusions the authors make are consistent with the evidence they provide.
Comment
Regarding Figure 5C. It would be useful to include an endothelial and/or adherens junction marker during the preparation of the immunofluorescence histology. At the moment there is no clear marker that delineates the vascular from the extra-vascular space making it difficult to interpret the distribution of aquaporin-4, in particular. In addition, the punctate distribution of albumen is rather strange. Wouldn’t it be distributed in the same way as aquaporin-4?
